# Patient-Reported Quality of Care for Osteoarthritis in General Practice in South Tyrol, Italy: Protocol for Translation, Validation and Assessment of the OsteoArthritis Quality Indicator Questionnaire (OA-QI)

**DOI:** 10.3390/mps6020028

**Published:** 2023-03-10

**Authors:** Christian J. Wiedermann, Pasqualina Marino, Antje van der Zee-Neuen, Isabella Mastrobuono, Angelika Mahlknecht, Verena Barbieri, Sonja Wildburger, Julia Fuchs, Alessandra Capici, Giuliano Piccoliori, Adolf Engl, Nina Østerås, Markus Ritter

**Affiliations:** 1Institute of General Practice and Public Health, Claudiana College of Health Professions, 39100 Bolzano (BZ), Italy; 2Department of Public Health, Medical Decision Making and Health Technology Assessment, University of Health Sciences, Medical Informatics and Technology, 6060 Hall, Austria; 3Center for Physiology, Pathophysiology and Biophysics, Institute of Physiology and Pathophysiology, 5020 Salzburg, Austria; 4Gastein Research Institute, Paracelsus Medical University, 5020 Salzburg, Austria; 5Institute of Nursing Science and Practice, Paracelsus Medical University, 5020 Salzburg, Austria; 6Department of General Medicine, South Tyrolean Health Care Service, 39100 Bolzano (BZ), Italy; 7Center for Treatment of Rheumatic and Musculoskeletal Diseases (REMEDY), Diakonhjemmet Hospital, N-0319 Oslo, Norway; 8Ludwig Boltzmann Institute for Arthritis and Rehabilitation, Paracelsus Medical University, 5020 Salzburg, Austria; 9School of Medical Sciences, Kathmandu University, Dhulikhel 45200, Nepal

**Keywords:** osteoarthritis, quality indicator, quality of care, German, Italian, measurement properties

## Abstract

Background: Evidence-based recommendations for the treatment of knee and hip osteoarthritis are similar internationally. Nevertheless, clinical practice varies across countries. Instruments for measuring quality have been developed to improve health care through targeted interventions. Studies on health service quality must consider the structural and cultural characteristics of countries, because each of their strengths and weaknesses differ. However, such instruments for health-related patient-reported outcomes for osteoarthritis have not yet been validated in German and Italian languages. Objectives: In order to be able to set targeted measures for the improvement of prevention and non-surgical treatment of osteoarthritis in South Tyrol, Italy, the quality of care must be recorded. Therefore, the aim of the project is to update, translate, and validate the OsteoArthritis Quality Indicator (OA-QI) questionnaire version 2, an established and validated questionnaire in Norwegian and English, for Germany and Italy. The second aim is to determine the quality of care for osteoarthritis of the hip and knee in a sample of patients who consult general practice in South Tyrol, and for comparison with patients who are admitted to rehabilitative spa-treatments for osteoarthritis in the state of Salzburg, Austria. Discussion: The results of this study will enable the identification and closure of gaps in osteoarthritis care. Although it is expected that body weight and exercise will play special roles, other areas of nonsurgical care might also be involved.

## 1. Introduction

Osteoarthritis (OA) is a joint disease characterized by joint stiffness, pain, disability, and reduced quality of life and is a major cause of pain and disability in the adult population worldwide [1]. For disease diagnosis, radiologic imaging is required only in cases in which the diagnosis is unclear or surgical treatment is necessary; otherwise, the diagnosis is made according to clinical criteria [2,3]. The prevalence of OA increases with age; almost one in two people will develop symptomatic knee OA and one in four will develop symptomatic hip OA during their lifetime [4,5,6,7]. OA exacerbates physical inactivity, which is partly responsible for a number of physical and psychological consequences that increase the risk of morbidity and mortality [8].

With an aging population and obesity epidemic, the prevalence of OA is expected to increase substantially. Based on a recent projection, the overall increase in the total number of patients with OA from 2019 to 2080 is expected to be 38% for both men and women. The most affected groups were those aged 70–79 and 80 or more years. The increases based on the assumed main scenario (mean fertility, rate of immigration, and life expectancy) are forecasted to be 45% and 245% for men and 28% and 148% for women. Assuming a more plausible population growth scenario (higher fertility and rate of immigration, longer life expectancy), these numbers are 74% and 360% (men) and 48% and 209% (women), respectively [6]. In light of this enormous increase in OA incidence, it is likely that this disease will lead to a substantial socioeconomic burden on healthcare systems in the near and far future. These findings will lead to the development of sustainable strategies for the treatment and prevention of OA.

OA, especially of the knee and hip, affects patients’ quality of life and is a major challenge for the healthcare system. Guidelines for the prevention and treatment of OA recommend a biopsychosocial approach, in which general practice and rehabilitation play special roles. Evidence-based recommendations and standards for OA management have been defined and have remained essentially unchanged for over a decade [9,10,11,12]. These recommendations, which include (i) patient education, (ii) self-management, (iii) exercise, and (iv) weight reduction, are beneficial for reducing pain and improving functionality. National quality registers were established years ago in various countries [13]. Whether these have led to measurable improvements in the disease burden remains unclear in many cases. Despite these benefits and the relative ease of implementation, they are often not offered to patients with symptomatic OA or implemented in clinical care [14,15,16].

### 1.1. Quality of Osteoarthritis Care in General Practice

Studies on the treatment quality of OA in Northern European countries indicate that quality can be significantly improved in different care settings [17]. To maximize the benefits of OA care, it is important to implement evidence-based and cost-effective care and reduce the use of treatments with limited or no evidence, while reducing the use of resources. Previous research has shown that family physicians (GPs) are reluctant to talk to their patients about relevant psychosocial issues and body weight [18]. It hasalso been shown that GPs favor monitoring patients’ physical function, pain, and analgesia over body mass index (BMI), self-management plans, and exercise advice [19]. Indeed, some GPs feel that they have insufficient expertise to advise patients about exercise [20].

A small number of best-practice initiatives to improve the quality of OA care have been carried out with varying results [13]. However, patient information and exercise have been identified as core treatments for OA with the potential to be improved through self-management programs [21]. Therefore, it is important to assess the quality of OA care, including these two aspects of practice, before implementing regional programs [22]. A recent analysis of quality indicators (QIs) showed large heterogeneities between healthcare systems in terms of exercise therapy, weight counseling, and referrals for laboratory and imaging tests [23]. These differences highlight the need for healthcare systems to carefully select QIs for knee and hip OA to validate the quality of OA care. It is strongly recommended that QIs be reviewed against the most recent guidelines before they are implemented.

### 1.2. Quality Indicators (QIs) of Osteoarthritis Care

QIs can be used to assess healthcare quality. These indicators can refer to measurable elements of the metrics of material and human resources of healthcare (i.e., the structures), the activities performed (i.e., the process), and the changes in health status resulting from the healthcare provided (i.e., the outcomes) [24]. QI sets developed from OA care recommendations can be used to monitor and assess the quality of care provided. A systematic review of QI studies on knee and hip OA treatment concluded that QI sets are heterogeneous, precluding cross-cultural use and international comparisons, and only a few studies have included patient perspectives [23]. Patient-reported quality of care showed a large variation in different quality indicators across four European countries, possibly reflecting differences in healthcare priorities [17].

#### OsteoArthritis Quality Indicator (OA-QI) Questionnaire

The OsteoArthritis Quality Indicator (OA-QI) questionnaire was developed in 2010 to measure patient-reported health-related quality of OA care [25]. The items of the instrument were based on published QIs from the literature and further refined through expert panels and patient interviews. Content validity was assessed as satisfactory after the OA-QI items were rated as relevant by the patient research partners and expert panels. The OA-QI was revised in 2015 [26]. The concept of the construct is for the disease specificity of care for OA and no other rheumatic diseases. The OA-QI was originally published in Norwegian and is available in Dutch [27], Danish, English, and Portuguese [17]. It has been successfully used for quality assessment in various settings in Denmark [28], Norway [29,30], Australia [31], and the United Kingdom [26].

The OA-QI questionnaire was the first validated instrument to measure patient-reported outcomes as a quality indicator for person-centered OA care [25]. It was revised in 2015 based on feedback (OA-QI v2) [30]. The number of items in the OA-QI v2 reduced from 17 to 16. The revised questionnaire was then completed. The questionnaire took three minutes to complete. The achievement of the QI items (i.e., the success rate for the answer options yes/no/not very concerned in a value range between 0 and 100) was calculated as a percentage (the total number of items achieved divided by the number of items eligible for each participant). A score of 100 indicates the best quality of care rating. The questionnaire is easy to use and recommended for use in primary care and general practice. Internal consistency, inter-observer reliability, and measurement errors were not tested. Reliability (intra-observer and test-retest) was tested and the intraclass correlation coefficient (ICC) was 0.89 (95% CI 0.83 to 0.93).

The instrument was tested on 13 individuals with OA, followed by a short interview to assess the comprehensibility of the questionnaire. Content validity was rated satisfactory. Structural validity was rated as acceptable based on six predefined hypotheses. To assess construct validity, hypothesis tests were conducted and all ten a priori hypotheses were confirmed. The cross-cultural validity of the translated OA-QI was also tested, and the instrument was used in national and international studies. The minimum significant difference (MSD) after participation in an OA patient education program was 20.4. This instrument can be used free of charge. The English version of the instrument is available in the original paper [30] but has not yet been validated in this language.

The OA-QI v2 consists of 16 self-administered items rating the individually perceived quality of care with selected responses (i.e., yes/no/not severely troubled), resulting in a total score ranging from 0 to 100 [30]. Higher scores in this range represent better quality of care. The reliability of the OA-QI v2 was estimated to be higher than that of the OA-QI v1 and its validity was acceptable. Therefore, the new version was recommended for future use as an outcome measure in studies to improve OA care [30]. The reliability, responsiveness, and interpretability of the OA-QI v2 were tested using the COSMIN checklist, which focuses on assessing the methodological quality of studies on the measurement properties of patient-related health outcomes with repeated evaluations [32].

### 1.3. Objectives

The main purpose of this study is to update, translate, and validate the OsteoArthritis Quality Indicator (OA-QI) questionnaire version 2, an established and validated questionnaire in Norwagian, for Germann and Italian languages, to assess the extent to which evidence-based treatment recommendations for OA care are followed at the regional level in South Tyrol, Italy, and to compare the survey results with those of a selected group of patients with OA in spa treatment for rehabilitation in Salzburg, Austria. To evaluate the quality of OA care, patients who contact their GP or seek spa-treatment because of complaints caused by OA of the hip or knee will be asked to fill out the OA-QI v2 questionnaire [30] that allows the quality of previous medical care to be assessed. The rehabilitation patient group is expected to have received a higher degree of attention with respect to OA as a (possible) cause of their complaints prior to admission to treatment. Therefore, this group may serve to disclose an attainable “standard” for optimized OA patient care in general practice.

As the OA-QI questionnaire is not yet available in German or Italian in a tested format, the questionnaire will be translated and culturally adapted to German- and Italian-speaking patients in the following steps: initial translations, synthesis of the translations, back translations, expert committee review, test of the pre-final versions, and development of the German and Italian versions of OA-QI v2 (G-OA-QI v2 and I-OA-QI v2, respectively). This phase will include testing by patient representatives.

In accordance with the original definition of OA quality, the individual QI items were based on the 2015 recommendations of professional societies for the treatment of knee and hip OA. After relevant changes in the treatment recommendations of the guidelines occurred recently, the third aim of the study is to control and eventually update the individual QI items of the G- and I-OA-QI v2. The resulting versions will be tested for validity in patients with OA in South Tyrol and Salzburg and finally used for a cross-sectional prospective observational quality assessment study.

## 2. Methods

### 2.1. Institutional Settings

Scientific collaboration between the Institute of General Practice and Public Health (IGPPH) at the College of Health Professions−Claudiana in Bolzano and the Paracelsus Medical University (PMU) is longstanding and focused on quality of care [33,34,35,36]. Indicators for assessing the quality of primary care for chronic diseases were compared between Salzburg and South Tyrol in a study performed by the IGPPH in Bolzano and the Institute of General Practice, Family Medicine, and Preventive Medicine of the PMU in Salzburg. In general practice, quality indicators were assessed for chronic conditions including diabetes mellitus type 2, hypertension, coronary heart disease, cerebrovascular disease, peripheral arterial disease, chronic heart failure, atrial fibrillation, and chronic obstructive pulmonary disease, but not knee or hip OA [37]. 

The Institute of Physiology and Pathophysiology of the PMU harbors the Gastein Research Institute and is a research unit at the Ludwig Boltzmann Institute for Arthritis and Rehabilitation. Recent studies have focused on the projection of the expected number of OA patients to provide a meaningful basis for policymakers when planning and budgeting efforts to treat and prevent OA [6]. 

### 2.2. Translation, Update and Validation of the English Version of the OsteoArthritis Quality Indicator (OA-QI) Questionnaire Version 2 into German and Italian

#### 2.2.1. Stepwise Translation Process

First, the authors of OA-QI v2 were contacted, and permission was obtained for translation into German and Italian. The authors of the OA-QI v2 also confirmed that a German or Italian version of the instrument has not yet been developed.

The English version of OA-QI v2 (Table 1) will be assessed for the need of cross-cultural adaption by a professional translation company and translated into Italian and German following an established forward-backward translation procedure, with independent translations and back translations. If cross-cultural adaption is deemed necessary by the translation experts, cognitive interviews to assess after-translation content validity from a patient perspective will be conducted with a limited number of participants (*n* = 5). Internal validity will be assessed for test-retest reliability using the intraclass correlation coefficients, agreement between assessments with Bland–Altman plots, and construct validity with Spearman’s correlation coefficients. Construct validity analyses will be performed using predefined hypotheses, as described in [25]. 

The English version of the OA-QI v2 will be translated by a translation company specialized in healthcare. Members of the research teams at the Institute of General Practice and Public Health (IGPPH) in Bolzano and the Paracelsus Medical University (PMU) in Salzburg then review the Italian and German translations.

To verify the accuracy of the translation and update the questionnaire items, the two documents will then be sent to two rheumatologists in Italy and Austria, whose suggestions for changes, if any, are incorporated. 

Ideally, sets of quality indicators should be updated frequently to reflect the current evidence-based treatment recommendations. The OA-QI v2 was updated in 2015–2016, then tested for measurement properties, and published in 2018 [30]. Since then, except for the European League Against Rheumatism (EULAR) [38], the Osteoarthritis Research Society International (OARSI) [11], American College of Rheumatology (ACR) [12], and National Institute for Health and Care Excellence (NICE) [39], treatment recommendations for paracetamol as first-line pharmacological treatment have changed. Hence, the items in the OA-QI v2 on this aspect, as well as topical or oral nonsteroidal anti-inflammatory drugs, should be updated. The proposal will be made to rheumatologists to update item #12 from ‘If you have joint pain, was paracetamol the first recommended medication?’ to ‘If you have joint pain, was paracetamol or a nonsteroidal anti-inflammatory drug the first medication that was recommended?’, and to update item #13 from ‘If you have prolonged severe joint pain, which is not relieved sufficiently by paracetamol, have you been offered stronger pain killing medications? (e.g., co-codamol, codeine, tramadol, co-proxamol, co-dydramol, dihydrocodeine)’ to ‘If you have prolonged severe joint pain, which is not relieved sufficiently by a nonsteroidal anti-inflammatory drug or paracetamol, have you been offered stronger pain killing medications? (e.g., co-codamol, codeine, tramadol, co-proxamol, co-dydramol, dihydrocodeine).

The two translated and reviewed G-OA-QI v2 and I-OA-QI v2 documents are then back-translated into the English language by the certified translation company, and the back-translated questionnaire is compared with the original OA-QI v2 to identify any major discrepancies (Figure 1).

#### 2.2.2. Pilot Survey of German and Italian OsteoArthritis Quality Indicator Version 2 Questionnaires in Patients with Knee and Hip Osteoarthritis

To test for clarity of the translated German and Italian G-OA-QI v2 and I-OA-QI v2 questionnaires and their validity, a pilot study will be conducted with 25 German-speaking and 25 Italian-speaking knee or hip osteoarthritis patients, respectively (for patient selection, see below). For this purpose, the patients will answer the 16 items of the G-OA-QI v2 and I-OA-QI v2 and provide information regarding selected socio-demographic and clinical characteristics. To calculate the test-retest reliability, they will fill out the respective questionnaires twice (i.e., at baseline and two weeks later) under the prerequisite that they do not see health professionals in the interim. Prior to this, the questionnaire items will be discussed with a subgroup of five patients per language in the context of cognitive debriefing interviews.

### 2.3. Participants Recruitment and Eligibility Criteria

Participants’ health-related reports on the quality of hip and knee OA care will be assessed in a cross-sectional survey using the validated Italian or German version of the updated OA-QI v2 questionnaire, according to the patient’s mother tongue, in a sample of 220 patients with OA in South Tyrolean general practices (50 for validation of the I- and G-OA-QI v2 questionnaires and 170 for quality assessment) and 150 patients with OA visiting rehabilitation facilities in the state of Salzburg for OA treatment (all for quality assessment). The sample size for the cohort was based on a pragmatic approach based on the number of referrals from patients with knee and hip OA during a 2-year inclusion period and is chosen to enable subgroup analyses above a minimal participant number of 50 each. Sample sizes for the validation phase were determined according to current scientific recommendation [41]. For the necessary number of cases for the quality assessment study, there are no uniform recommendations for the a priori calculation. We followed the subject-to-item ratio recommendation [42], which ranges from 1:5 to 1:30 in the literature. With 16 items of the used OA-QI v2 tools, the number of 170 patients in South Tyrol and 150 patients in Salzburg, which was set for pragmatic reasons (number of GPs and average number of patients in their outpatient clinics), corresponds to a ratio of about 1:10.

#### 2.3.1. General Practice

Subjects in South Tyrol are recruited in up to 25 GP practices of the Department of Basic Medical Services of the South Tyrolean Public Health Services. A pragmatic approach to the inclusion of participants based on the GPs’ diagnosis of knee or hip OA is applied, irrespective of the diagnostic criteria the GPs use. Patients presenting with unspecified symptoms or diagnoses, such as ‘knee or hip pain’ or ‘knee or hip problems’, will be considered for recruitment. The inclusion criteria defining OA diagnosis according to NICE are people who (i) are 45 years or older, (ii) have activity-related joint pain, and (iii) have either no morning joint-related stiffness or morning stiffness that lasts no longer than 30 min. Imaging to diagnose OA is not routinely used unless there are atypical features or features that suggest an alternative or additional diagnosis [39]. Current medications for OA (analgesics, nonsteroidal anti-inflammatory drugs, agents modifying the structure of connective tissue, and potentially disease-modifying OA drugs, intra-articular therapy, corticosteroids, visco-supplementation, and closed joint cleaning) will be documented. The exclusion criteria will be as follows: malignant illness, rheumatoid or other inflammatory arthritis, severe degeneration of the hip or knee joint (Kellgren and Lawrence Grade IV [43]), other inflammatory rheumatic diseases, mental or psychiatric disorders, inability to cooperate with the study requirements, and involvement in any other pharmaceutical or exercise studies at the moment.

#### 2.3.2. Rehabilitation Facilities in the Austrian State of Salzburg

Subjects in Austria are recruited in up to 25 GP practices and through spas and rehabilitation physicians prior to the initiation of treatment. The eligibility criteria are equal to those applied in the recruitment process in South Tyrol. 

In addition to direct recruitment by physicians as mentioned above, the Gastein Research Institute may collect relevant data by extending the already existing ‘Radon indication registry for the assessment of pain reduction, increase in quality of life, and improvement in body functionality through low-dose Radon hyperthermia therapy (RadReg)’ using the G-OA-QI v2 questionnaire [44]. This registry collects data from individuals visiting the valley of Gastein for spa-treatment, including low-dose radon for a variety of rheumatic diseases including OA. Registry subjects are recruited by physicians participating in treatment spa centers in the Gastein Valley. Therefore, these physicians are already trained in handling the RadReg questionnaires and will additionally receive free skills training to aid them in the recruitment of participants for the current study. 

No sample size calculation was performed, but post-hoc analyses will provide insights into the power of the study.

### 2.4. Quality Indicator, Demographic and Disease Charactieristic

The G-OA-QI v2 and I-OA-QI v2 questionnaires will be tested to assess the quality of OA care in the respective samples of consecutive OA patients participating in general practices in South Tyrol and in participating general practices, health care, and spa/rehabilitation centers in the state of Salzburg. As described in [30], a QI item will be considered achieved if the participant has checked ‘Yes.’ An item was considered “eligible” if the participant responded ‘Yes’ or ‘No’ for that item, whereas items were considered ‘not eligible’ and excluded from analysis if there was a missing/ambiguous response or if the participant had responded ‘Don’t remember,’ ‘Not overweight,’ ‘No such problems,’ and so on. Hence, the total number of eligible items varied across participants.

A total of 170 German or Italian speaking subjects and 150 German-speaking subjects will be tested in South Tyrol and Salzburg, respectively. In South Tyrol, patient responses to the questionnaires will be collected in general practice before the personal visit of the patient to the GP. In Austria, the assessment will be performed equally in the case of recruitment through GPs or immediately after the patients’ admission and before the start of their treatments in the case of recruitment through spa/rehabilitation centers.

The OA-QI v2 will be supplemented with demographic and clinically relevant data (Table 2) and will include the severity of OA (Lequesne Index [45] in its German [46] or Italian [47] versions) and EQ-5D-5L with subscales on mobility, self-care, usual activities, pain/discomfort, and anxiety/depression [48] for German [49] and Italian [50] in addition to the duration of knee or hip problems, other affected joints, and any surgical joint interventions. The Western Ontario and McMaster Universities (WOMAC) OA index subscale will be used in its Italian [51] and German [52] versions to assess physical function.

### 2.5. Study Registry Entry and Ethics

This study is registered in the ISRCTN registry [55]. The Scientific Ethics Committee of the Autonomous Province of Bolzano, Italy reviewed the study protocol and approved the study conduct on 20 October 2022 (No. 103-2022). Ethical approval will be obtained from the study center in Salzburg, Austria, according to the national regulations. The study will be conducted according to the standards of good clinical and scientific practice in compliance with the Declaration of Helsinki [56]. Furthermore, the guidelines for ‘Strengthening the Reporting of Observational studies in Epidemiology’ (STROBE) for the publication of observational studies will be followed [57]. 

### 2.6. Study Outcome Parameters

#### 2.6.1. Primary Outcome

Achievement of the QI items of the G-OA-QI v2 and I-OA-QI v2 tools by patients with knee and hip OA in general practice in South Tyrol and rehabilitative spa treatment will be the primary outcome parameters. The mean total pass rate will be calculated as a percentage, as described [30] for the whole sample, as well as for subgroups including type of OA, language, and treatment location.

#### 2.6.2. Secondary Outcomes

Differences within the same healthcare setting will be identified as secondary outcomes depending on demographic and clinical characteristics.

### 2.7. Trial and Data Management

The development and implementation of the study will follow the principles of the Declaration of Helsinki [56]. This type of data collection will be implemented by IGPPH in Bolzano and conducted in a pseudonymous form. Data provided by participants from online and transcribed paper questionnaires will be collected centrally in the SoSci Survey Software, version 3.2.46 (SoSci Survey GmbH, Munich, Germany). The online questionnaires are programmed such that all items have to be answered. The data are stored by IGPPH in Bolzano, Italy, and made available to the research team upon request after the end of the data analysis period. Data backup is regularly performed. Standard operating procedures (SOPs) chordate study procedures for various study assistants with appropriate training to regulate parallel procedures.

### 2.8. Statistical Analyses

Descriptive statistics will describe these data according to their metric properties, and regression analyses will be performed to explore the association between the questionnaire scores and predefined clinical outcomes. Statistical analyses will be performed using the software package IBM SPSS Statistics for Windows and STATA. The results of the study will have the ability to identify strengths and weaknesses in the quality of OA care in the two study cohorts of South Tyrol and Salzburg, and to determine the association between the quality of care and clinical outcomes. 

## 3. Discussion

General practice and primary care have become increasingly relevant in the care of OA patients. This study provides an overview of the quality of care for knee and hip OA after consulting a GP in South Tyrol and the state of Salzburg or spa/rehabilitation treatment in the state of Salzburg. The strength of this study is that patients are included consecutively from centers that represent both rural and urban areas of Northern Italy and the state of Salzburg, thus increasing the representativeness of the study population.

The use of self-report questionnaires containing retrospective information about previous treatments carries the risk of recall bias [58]. Another limitation is that it may include a small number of patients who do not meet the diagnostic criteria for OA, as the study is based on referrals from general practitioners for knee or hip OA from patients with non-specific diagnoses such as “knee pain” or “knee problems” if their age is ≥45 years. However, the self-report approach is the only way to collect information on patient-reported quality of care.

Risks for meeting preset milestones include the recruitment of GPs study sites and planned patient numbers in both general practice and participating health centers. As similar study protocols have been successfully completed in the past and various sites will participate in the recruitment of patients in the state of Salzburg, we are confident that the project goals will also be achieved.

This study was approved by the Italian Regional Ethical Committee of the Province of Alto Adige (No. 103-2022). Data will be anonymized and handled in line with the General Data Protection Regulation and Italian Data Protection Act. The study results will be submitted to international, open-access, peer-reviewed journals and disseminated at conferences. The validated translation of the OA-QI v2 into Italian and German is expected to result in two open-access peer-reviewed publications. The original article on the quality of OA care will be published in a clinical rheumatology journal and is expected to propose specific interventions for quality improvement in both South Tyrol and Salzburg.

## 4. Conclusions

The results of this study will enable the identification and closure of gaps in OA care. Although it is expected that body weight and exercise will play special roles, other areas of nonsurgical care might also be involved.

## Figures and Tables

**Figure 1 mps-06-00028-f001:**
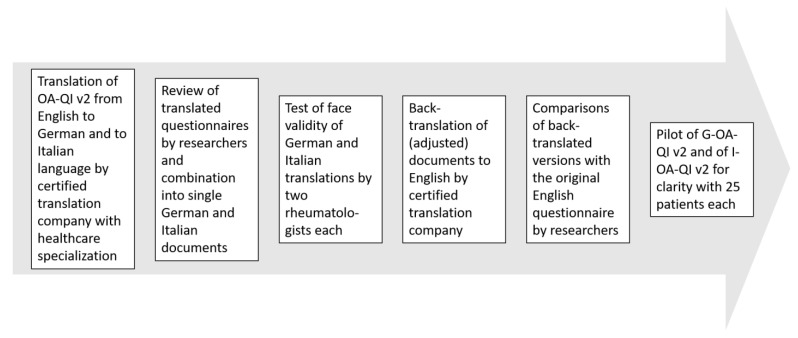
Step-by-step translation of the OsteoArthritis Quality Indicator version 2 questionnaire. Abbreviations: OA-QI v2 = OsteoArthritis Quality Indicator version 2; G-OA-QI v2 = German OA-QI v2; I-OA-QI v2 = Italian OA-QI v2. Reproduced with modification from Omair et al. [40] under a Creative Commons Attribution—NonCommercial (unported, v3.0) License (http://creativecommons.org/licenses/by-nc/3.0/, accessed on 31 December 2022). Copyright © 2021, The Authors. This reuse has not been endorsed by the licensor. The source reference is “[40]” and is available at https://www.ncbi.nlm.nih.gov/pmc/articles/PMC8253897/, accessed on 31 December 2022.

**Table 1 mps-06-00028-t001:** English translation of the OsteoArthritis Quality Indicator version 2 (OA-QI v2) [30].

Questions on the Treatment of Your Osteoarthritis				
There are several different treatment alternatives for osteoarthritis. What treatment, information or advice have you received from health professionals for your osteoarthritis in the past year ^†^? For each question, please cross off one of the boxes provided				
		Yes	No	Don’t remember
1	Have you been given information about osteoarthritis from a health professional?	□	□	□
2	Have you been given information about different treatment alternatives?	□	□	□
3	Have you been given information about how you can self-manage the disease?	□	□	□
4	Have you been given information about the importance of physical activity and exercise?	□	□	□
5	Have you been referred or offered a referral to a health professional who can advise you about physical activity and exercise?	□	□	□
		Yes	No	No overweight
6	Have you been advised to lose weight, if you are overweight?	□	□	□
7	Have you been referred or offered a referral to someone who can help you to lose weight, if you are overweight?	□	□	□
		Yes	No	No such problems
8	If you have problems with daily activities, have these problems been assessed by a health professional?	□	□	□
9	If you have problems with walking, has your need for a walking aid been assessed? (e.g., stick, crutch or walker)	□	□	□
10	If you have problems related to other daily activities, has your need for appliances and aids been assessed? (e.g., splints, assistive technology for cooking or personal hygiene, a special chair)	□	□	□
		Yes	No	No pain
11	If you have joint pain, has it been assessed by a health professional?	□	□	□
12	If you have joint pain, was paracetamol the first medication that was recommended?	□	□	□
		Yes	No	No prolonged severe pain
13	If you have prolonged severe joint pain, which is not relieved sufficiently by paracetamol, have you been offered stronger pain killing medications? (e.g., co-codamol, codeine, tramadol, co-proxamol, co-dydramol, dihydrocodeine) *	□	□	□
		Yes	No	Not taking such drugs
14	If you use anti-inflammatory medications, have you been given information about the effects and possible side-effects of this medication? (e.g., ibuprofen (Nurofen, Brufen), diclofenac (Voltarol), naproxen (Naprosyn), celecoxib (Celebrex)) *	□	□	□
		Yes	No	Not experienced such deterioration
15	If you have experienced an acute deterioration of your symptoms, have you been given or offered a steroid injection?	□	□	□
		Yes	No	Not severely troubled
16	If you are severely troubled by your osteoarthritis, and exercise and medication do not help, have you been referred or offered a referral for an assessment for operation? (e.g., joint replacement)	□	□	□

^†^ One year was chosen as the optimal timeframe. * Drug trade name examples will be adapted to patients’ countries.

**Table 2 mps-06-00028-t002:** Demographic and osteoarthritis disease characteristics items to be collected and response categories. Reproduced with modification from Darlow et al. [53] under an Attribution-NonCommercial-NoDerivatives 4.0 International (CC BY-NC-ND 4.0) License (https://creativecommons.org/licenses/by-nc-nd/4.0/, accessed on 2 January 2023). Copyright © 2021, The Authors. This reuse has not been endorsed by the licensor. The source reference is “[53]” and is available at https://pubmed.ncbi.nlm.nih.gov/36474995/, accessed on 2 January 2023.

Item	Response Categories
Birth year	Year (1900–2002)
Gender	Male
	Female
	Gender diverse
	Prefer not to answer
Ethnicity	Free text (no response option framework appropriate for all countries)
Native language	Italian
	German
	Other (specify)
Country of residence	Italy
	Austria
	Other (specify)
Socioeconomic circumstance ^1^	1 Not at all difficult
	2
	3
	4
	5 Extremely difficult
Rurality	Urban
	Rural 1 (25–60 min travel to urban centre of 30,000 people or more)
	Rural 2 (60–90 min travel to urban centre of 30,000 people or more)
	Rural 3 (>90 min travel to urban centre of 30,000 people or more)
Highest level of education	Some secondary education (high school)
	Completed secondary education (graduated high school)
	Trade/technical/vocational training
	Some undergraduate education (college or university)
	Completed undergraduate education (college or university)
	Some postgraduate education
	Completed postgraduate education (masters or doctorate)
	Other (please specify)
Occupation	Manager
	Professional
	Technician or Trades Worker
	Community or Personal Service Worker
	Clerical or Administrative Worker
	Sales Worker
	Machinery Operator or Driver
	Labourer
	Homeworker
	Unemployed looking for work
	Unemployed not looking for work
	Student
	Retired
	Unable to work due to health reasons
	Other (please specify)
Pain duration	Less than one year
	One to two years
	Two to five years
	Five to ten years
	Ten to fifteen years
	Fifteen to twenty years
	More than twenty years
Diagnosis of OA by health professional (multiple options may be selected)	Nil
	Left hip
	Right hip
	Left knee
	Right knee
Joint replacement and year (multiple options may be selected)	Nil
	Left hip
	Right hip
	Left knee
	Right knee
Where received OA information (multiple options may be selected)	No information received
	GP or family doctor
	Surgeon
	Another doctor (such as sports doctor or rheumatologist)
	Nurse
	Physiotherapist or physical therapist
	Osteopath
	Chiropractor
	OA rehabilitation programme
	Arthritis educator
	Arthritis support group
	Other people with OA
	Family or friends
	Internet/website
	Television
	Information booklets
	Other (please specify)

^1^ Survey questions of the OECD INFE financial literacy core questionnaire [54].

## Data Availability

The data presented in this study will be available upon request from the corresponding author. The data are not publicly available for language and ethnicity reasons in the politically autonomous state of the Italian region, Trentino—Alto Adige.

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
