# Peer review of "Patient-Reported Quality of Care for Osteoarthritis in General Practice in South Tyrol, Italy: Protocol for Translation, Validation and Assessment of the OsteoArthritis Quality Indicator Questionnaire (OA-QI)"

_mps, 2023, doi:10.3390/mps6020028_

Round 1

Reviewer 1 Report

the authors is studying a measurement tool for the high prevalence of disease which affect million of patients. 

The OsteoArthritis Quality Indicator (OA-QI) questionnaire was developed to measure patient-reported health-related quality of OA care, has been widely used in Europe. the reliable version for Italy and Germen patients are needed. it is why the study is important.

the authors did state that to measure patient-reported health-related quality of OA care. it may be helpful for readers to understand that they can explain how they get  the number of GP and number of patients. 

Reviewer 2 Report

Comments

General: The present manuscript “Patient-reported quality of care for osteoarthritis in general practice in South Tyrol, Italy: A study protocol by  Christian J. Wiedermann et al.

Specific:

1.      The title of the manuscript should be modified – should be specific

  1. The materials which were used are not given in details.
  2. Introduction should be brief and to the points.
  3. Results are poorly presented. Biochemical assays should be given in details,
  4. Enzymes or biochemical test should be done in relation to the osteoarthritis.
  5. The conclusion should be in little bit in details.

Conclusion:

It is a good work. The above paper may be accepted as after attempting the above queries
